# Deciphering the Antibacterial Mechanisms of 5-Fluorouracil in *Escherichia coli* through Biochemical and Transcriptomic Analyses

**DOI:** 10.3390/antibiotics13060528

**Published:** 2024-06-05

**Authors:** Muchen Zhang, Huangwei Song, Siyuan Yang, Yan Zhang, Yunrui Tian, Yang Wang, Dejun Liu

**Affiliations:** 1National Key Laboratory of Veterinary Public Health Security, College of Veterinary Medicine, China Agricultural University, Beijing 100193, China; zhangmuchen@cau.edu.cn (M.Z.); yangsiyuan@cau.edu.cn (S.Y.); b20233050453@cau.edu.cn (Y.Z.); s20233050987@cau.edu.cn (Y.T.); 2Guangdong Laboratory for Lingnan Modern Agriculture, Guangzhou 510642, China; 3School of Pharmaceutical Science and Technology, Hangzhou Institute for Advanced Study, University of Chinese Academy of Sciences, Hangzhou 310024, China; hwei_song@126.com

**Keywords:** 5-fluorouracil, drug repurposing, antibacterial mechanism, bacteria metabolism

## Abstract

The emergence of carbapenem-resistant Gram-negative pathogens presents a clinical challenge in infection treatment, prompting the repurposing of existing drugs as an essential strategy to address this crisis. Although the anticancer drug 5-fluorouracil (5-FU) has been recognized for its antibacterial properties, its mechanisms are not fully understood. Here, we found that the minimal inhibitory concentration (MIC) of 5-FU against *Escherichia coli* was 32–64 µg/mL, including strains carrying *bla*_NDM-5_, which confers resistance to carbapenems. We further elucidated the antibacterial mechanism of 5-FU against *E. coli* by using genetic and biochemical analyses. We revealed that the mutation of uracil phosphoribosyltransferase-encoding gene *upp* increased the MIC of 5-FU against *E. coli* by 32-fold, indicating the role of the *upp* gene in 5-FU resistance. Additionally, transcriptomic analysis of *E. coli* treated with 5-FU at 8 µg/mL and 32 µg/mL identified 602 and 1082 differentially expressed genes involved in carbon and nucleic acid metabolism, DNA replication, and repair pathways. The biochemical assays showed that 5-FU induced bacterial DNA damage, significantly increased intracellular ATP levels and the NAD^+^/NADH ratio, and promoted reactive oxygen species (ROS) production. These findings suggested that 5-FU may exert antibacterial effects on *E. coli* through multiple pathways, laying the groundwork for its further development as a therapeutic candidate against carbapenem-resistant bacterial infections.

## 1. Introduction

Only over 90 years after the discovery of antibiotics, the development and spread of antibiotic resistance among pathogens have limited the therapeutic effects of these drugs [1,2]. Among these antibiotics, carbapenems are considered as the critically important antibiotics to address bacterial infections. However, the emergence of carbapenem-resistant Gram-negative bacteria poses a severe threat to animal and human health, necessitating effective measures to address this problem [3]. Given the slow pace of new antibiotic development, repurposing existing drugs as antibiotics has become a key strategy in combating this crisis [4,5]. This approach has several advantages, including circumventing the high costs and lengthy timelines associated with developing new antimicrobial agents, as well as leveraging compounds with known pharmacological properties [6,7]. 

Nucleoside and nucleobase analogue drugs structurally resemble the nucleosides and nucleotides found in the cell, and they can participate in and disrupt the synthesis of DNA and RNA, thereby blocking cell division [8,9]. Recent studies have revealed that these drugs have anticancer and antibacterial effects [10]. Therefore, re-utilizing these drugs and exploring new uses for these compounds is a feasible strategy to address the current issue of clinical bacterial resistance. One of the nucleobase analogue drugs, 5-fluorouracil (5-FU), is primarily known for its role as an anticancer drug. Its mechanism in tumor cells involves the inhibition of thymidylate synthase, thereby inhibiting DNA synthesis [11,12]. In addition, 5-FU exhibits antibacterial effects on bacteria including *E. coli*, *Staphylococcus aureus*, and *Streptococcus suis* [13]. It can inhibit the formation of biofilms and reduce bacterial virulence in *E. coli* and *Pseudomonas aeruginosa* [14,15,16,17]. Moreover, central venous catheters externally coated with 5-FU represent a safe and effective alternative to catheters externally coated with chlorhexidine and silver sulfadiazine when used in critically ill patients [18]. Similar to its action in cancer cells, 5-FU inhibits thymidylate synthase in *E. coli*, thereby inhibiting bacterial growth and reproduction [9,19]. However, the bactericidal mechanisms of 5-FU, particularly its effects on bacterial metabolism, have not been fully elucidated. Therefore, this study found that the MIC of 5-FU against *E. coli* carrying *bla*_NDM-5_ was 32–64 µg/mL and used a combination of biochemical assays and transcriptomic techniques to elucidate the resistant and antibacterial mechanisms of 5-FU in *E. coli*, thereby offering a novel approach to tackle the antibiotic resistance.

## 2. Results

### 2.1. Mutation in upp Confers Resistance to 5-FU

Initially, we examined the antibacterial activity of 5-FU, discovering that its MIC against *E. coli* ranged from 32 to 64 µg/mL, including strains carrying carbapenem resistance gene *bla*_NDM-5_ (Appendix A). In order to further investigate the antibacterial mechanism of 5-FU against *E. coli*, four strains (two *bla*_NDM-5_ positive *E. coli*, 15NN1 and 1DM10, and two standard *E. coli*, BW25113 and DH5α) were involved in serial passage assays to select resistant mutant strains to 5-FU, as shown in Figure 1. After continuous passage for 30 days, we selected the 5-FU-resistant mutant strains at MIC of 2048 µg/mL, which showed a 32-fold increase over the original strain (Figure 2a and Appendix A). Subsequently, a comparison of the growth patterns between the original and resistant strains revealed no significant differences in their growth conditions (Appendix A). Whole-genome sequencing and a single nucleotide polymorphism (SNP) analysis showed that the resistant strains harbored 3–4 nonsynonymous mutations, and three strains exhibited point mutations in the *upp* gene, which encodes uracil phosphoribosyltransferase (UPRT), resulting in either the premature termination of translation or the production of different amino acids in the *upp* gene (Table 1), indicating that the loss of UPRT function appeared to account for the 5-FU-resistance phenotype. 

To further confirm whether the *upp* gene could mediate bacterial resistance to 5-FU, we knocked out and complemented the *upp* gene in the standard strain BW25113 carrying the IncX3 plasmid, and a significant increase in MIC values (32 to 1024 µg/mL) was observed in the knockout strain (Figure 2a and Appendix A). Nevertheless, the knockout strain exhibited similar growth curves to the original strain (Figure 2b). In contrast, the complementation strain with the *upp* gene showed no change in its susceptibility to 5-FU (MIC=32 µg/mL). However, the growth of the complemented strain was slower, likely due to the high adaptive cost caused by the high-copy plasmid used for *upp* gene complementation (Figure 2b). These findings substantiate that the loss function of the *upp* gene would confer resistance to 5-FU in *E. coli* strains.

### 2.2. Transcriptomics Profile of 5-FU Treatment

To explore the antibacterial mechanisms of 5-FU, we performed a transcriptional analysis of *E. coli* 15NN1 after exposure to 5-FU at a concentration of 8 µg/mL (F8 Group) or 32 µg/mL (F32 Group). Raw data obtained from Illumina sequencing and clean reads were obtained after removing low-quality reads (Appendix A). Subsequently, all the clean reads were mapped to the reference genome of *E. coli* using HISAT2 software, and the total mapping ratio ranged from 99.56 to 99.67% (Appendix A). The mRNA expression exhibited high repeatability with correlation coefficients of more than 0.96 within the biological triplicates of each group (Appendix A). These findings indicated that high-quality sequencing data were obtained and suitable for further analysis. 

A comparative transcriptomic analysis was conducted to understand the gene expression variations between 5-FU-treated and untreated groups. Compared to the control group, the addition of 8 µg/mL 5-FU resulted in 602 differentially expressed genes (DEGs), including 288 upregulated genes and 374 downregulated genes (Figure 3a). The addition of 32 µg/mL 5-FU resulted in 1082 DEGs, including 460 upregulated genes and 622 downregulated genes (Figure 3d). A Gene Ontology (GO) enrichment analysis categorizes genes based on their functions into three groups: biological process, cellular component, and molecular function (Figure 3b,e). Among biological process, DEGs primarily participated in oxidation–reduction processes, cellular biosynthesis and metabolic process. In the cellular component, DEGs were mainly associated with protein-containing complexes and periplasmic space. Additionally, DEGs were primarily involved in molecular functions such as oxidoreductase activity, nucleic acid binding activity, and structural molecule activity. In a Kyoto Encyclopedia of Genes and Genomes (KEGG) enrichment analysis, both the different concentrations of 5-FU treatment groups resulted in DEGs associated with pathways such as pyruvate metabolism, pentose phosphate pathway, glycolysis, and ribosome biosynthesis. Additionally, DEGs in the F8 group were also involved in metabolic pathways such as purine metabolism and DNA repair and recombination protein (Figure 3c), while those in the F32 group participated in metabolic pathways including the TCA cycle, oxidative phosphorylation, and DNA replication (Figure 3f). 

Based on these results, we further focused on the expression levels of these specific genes involved in bacterial metabolism pathways. Specifically, we found that all genes in pyruvate metabolism were upregulated in 5-FU treatment groups except for *pta*, which was regulated and accounts for phosphate acetyltransferase (Figure 3h). The relative expression levels of DNA repair and combination and TCA cycle-related genes were also significantly increased (Figure 3g,i). In addition, an RT-qPCR assay was used to determine the expression of DEGs with the same RNA samples to validate the RNA-Seq data. All six DEGs (*rnfB*, *deaD*, *mgtA*, *murE*, *tdcB*, and *ackA*) were randomly selected and tested, and the RT-qPCR results were generally accordance with transcriptomic data (Appendix A).

### 2.3. 5-FU Induced DNA Damage and Enhanced Bacteria Metabolism

Previous research has shown that 5-FU can be metabolized intracellularly into fluorouridine monophosphate and fluorodeoxyuridine monophosphate, thereby inhibiting DNA synthesis [10]. Consequently, we utilized an ELISA kit to detect the DNA damage marker 8-hydroxydeoxyguanosine (8-OHdG) to investigate the effects of 5-FU on bacterial DNA damage. The results indicated a concentration-dependent increase in bacterial intracellular levels of 8-OHdG, suggesting that 5-FU can induce bacterial DNA damage (Figure 4a). To further assess whether the incorporation of 5-FU into DNA triggers a genotoxic stress response, we examined the effect of 5-FU treatment on the expression of the DNA-damage-responsive gene *recA* in *E. coli* by RT-qPCR (Figure 4b). The *recA* gene was significantly upregulated after 5-FU treatment in a concentration-dependent manner. However, when the resistant (15NN1-R) and knockout strain (BW25113-IncX3 ∆*upp*) were treated with 5-FU, no significant DNA damage was observed (Figure 4c). This result further supports the association between the presence of the *upp* gene and the DNA damage mechanism induced by 5-FU.

As uracil can enhance ATP levels in *S. aureus* [20] and 5-FU is a derivative of uracil, whether 5-FU also affects bacterial energy metabolism levels remains unclear. Transcriptomic studies have shown that 5-FU can disturb bacterial energy metabolism pathways, including the TCA cycle and the pyruvate metabolism. Therefore, we explored the impact of 5-FU on the intracellular ATP levels in *E. coli*. Indeed, compared to the untreated group, the original strain (15NN1 and BW25113-IncX3) showed an increase in intracellular ATP levels under 5-FU treatment. And the resistant and knockout strains also exhibited a significant increase in ATP levels, although the rise was not as pronounced as in the original strain (Figure 4d). These results confirmed that 5-FU not only leads to DNA damage, but also enhances bacterial metabolic levels. Bacterial oxidative phosphorylation is the primary pathway for ATP production, and the bacterial electron transport chain (ETC) plays a crucial role in bacterial respiration [21,22]. Considering that NADH is an important electron donor in the ETC [23], we assessed whether 5-FU can enhance bacterial respiration and found that 5-FU increased the NAD^+^/NADH ratio in both the original and knockout strains, though the increase was less pronounced in the knockout strain (Figure 4e). The imbalance of NAD^+^/NADH ratio can lead to the production of reactive oxygen species (ROS) [24,25]. Using the ROS probe 2′,7′-dichlorofluorescein diacetate (DCFH-DA), we observed that 5-FU markedly increased ROS generation in the original, resistant, and knockout strains, thereby ROS plays a key role in the antimicrobial mechanism of 5-FU (Figure 4f). Additionally, we tested the antibacterial effect of 5-FU on bacteria under different conditions. The results indicated that 5-FU significantly reduced the bacterial loads in exponentially growing cultures (Appendix A). However, the bactericidal effect of 5-FU was inhibited in stationary phase bacteria, as well as in exponential phase bacteria at 0 °C for *E. coli* 15NN1 (Appendix A). These findings further demonstrated that the impact of bacterial metabolic state on the antibacterial efficacy of 5-FU.

## 3. Discussion

The discovery of antibiotics is undoubtedly one of the greatest achievements in medicine and proved to be a turning point in human history by saving countless lives and controlling infectious diseases. However, the emergence of drug-resistant bacteria and the slow pace of antibiotic discovery have significantly affected clinical infection treatments [26,27]. Thus, there is an urgent need for new antibacterial drugs against the development of resistance. Drug repurposing is a cost-effective strategy to accelerate the drug development process [28]. Several nucleoside and nucleobase analogue drugs have already been explored for their potential as antibacterial agents [29]. 5-FU, an uracil derivative replacing hydrogen at position 5 with fluorine, exhibits both antibacterial and anticancer properties [30]. Previous research has showed that bacteria can metabolize 5-FU into 5-fluorodeoxyuridine monophosphate, which inhibits thymidylate synthase activity and blocks bacterial DNA replication, thus exerting its antimicrobial effects in *E. coli* [11]. However, the underlying mechanisms of 5-FU against *E. coli* remain unclear. Here we found that the *upp* gene can mediate resistance to 5-FU in *E. coli*. The UPRT, encoded by the *upp* gene, is widely present in prokaryotes and some lower eukaryotic cells and catalyzes the reaction of uracil with phosphoribosyl pyrophosphate to form uridine monophosphate and pyrophosphate, participating in the salvage synthesis pathway of bacterial pyrimidine metabolism [31]. Mutations in the *upp* gene inhibit this conversion process, leading to resistance by preventing the formation of the active metabolite (5-fluorouridine monophosphate). Except for the *upp* gene, a mutation in uridine monophosphate kinase-encoding gene *pyrH* gene in *E. coli* 15NN1, which can prevent the further conversion of 5-fluorouridine monophosphate and lead to resistance. Similarly, Singh et al. found that resistance of *Mycobacterium tuberculosis* to 5-FU is associated with mutations in *upp* or *pyrR* [9]. Additionally, previous research suggested that 5-FU exerts its antimicrobial action by inhibiting DNA synthesis [19]. Our study further revealed that 5-FU can cause significant bacterial DNA damage and trigger DNA repair by measuring DNA damage markers such as 8-OHdG and the transcription levels of *recA*. However, when the knockout strain (BW25113-IncX3 ∆*upp*) was treated with 5-FU, no significant DNA damage was observed. These findings collectively supported the association between the presence of the *upp* gene and the DNA damage mechanism induced by 5-FU.

With the development of high-throughput sequencing technologies, transcriptomic sequencing has played a significant role in studying the antimicrobial mechanisms of drugs [32,33]. Here, our transcriptional profiles revealed that the DEGs were primarily involved in bacterial energy metabolism, pyrimidine metabolism, DNA replication, and repair by comparing 5-FU treated groups with control group. The disruption of the pyrimidine metabolism pathway affects the synthesis of bacterial nucleotides, consistent with previous studies, suggesting that 5-FU can exert antibacterial effects by affecting nucleotide synthesis and interfering with nucleotide metabolism pathways [9]. Furthermore, Lopatkin et al. showed that variations in the expression of genes associated with central carbon and energy metabolism can influence the impact of antibiotics by affecting basic respiratory rates and tricarboxylic acid cycle activity [34]. The transcriptomic data showed that multiple DEGs were enriched in bacterial energy metabolism, such as the TCA cycle and pyruvate metabolism, aligning with results showing increased bacterial ATP levels and NAD^+^/NADH following 5-FU treatment, further indicating that 5-FU can also exert its antimicrobial effect by affecting bacterial energy metabolism. 

The metabolic state of bacteria affects their antibiotic susceptibility, and researchers have proposed a relationship between bacterial metabolism and antibiotic efficacy: (1) antibiotics alter the metabolic state of bacteria, contributing to bacterial death; (2) the metabolic state of bacteria influences their susceptibility to antibiotics; and (3) antibiotic efficacy can be enhanced by altering the metabolic state of bacteria [35,36]. In our study, we characterized the overall bacterial metabolic changes by measuring the biochemical indicators after 5-FU treatment and found that 5-FU significantly increased intracellular ATP levels, NAD^+^/NADH ratio, and ROS production in bacteria. Additionally, when the *upp* gene was knocked out, 5-FU treated bacteria showed no significant changes in the DNA damage marker 8-OHdG, but ATP and ROS levels still increased, indicating that 5-FU not only leads to DNA damage but also enhances bacterial metabolic levels. Furthermore, Yang et al. utilized a white-box machine learning approach to demonstrate that pyrimidine supplementation inhibits pyrimidine biosynthesis while promoting purine biosynthesis activity and ATP demand, thereby increasing antibiotic lethality [37]. This indicates that 5-FU has the potential to become an antibiotic adjuvant.

## 4. Materials and Methods

### 4.1. Bacterial Strains and Chemical Reagents

All the bacterial strains used in this study are listed in Appendix A. Unless otherwise noted, all strains were cultivated in LB broth or on LB agar (LBA) plates for the following tests. 5-FU were purchased from Aladdin (Shanghai, China).

### 4.2. Growth Curves of Bacteria

Briefly, overnight grown bacteria were diluted 1:100 into fresh LB at 37 °C for 4 h. The cultures were diluted to 10^6^ CFUs/mL and divided into a 96-well plate. The bacteria were cultured for up to 24 h, and growth curves were recorded under the wavelength of 600 nm with an interval of 1 h at 37 °C, using an Infinite M200 Microplate reader (Tecan, Männedorf, Switzerland). All the experiments were performed with at least three biological replicates.

### 4.3. Serial Passaging Assay to Evolve Resistance

Initially, a parent culture was prepared and utilized to inoculate a serially diluted series of 5-FU. Following incubation, the MICs of three biological replicates of both meropenem-resistant and meropenem-sensitive *E. coli* strains were determined against 5-FU.

Subsequently, the culture with sub-MIC concentration was used to inoculate a second serially diluted series of 5-FU and incubated overnight. The MIC was then determined the next day, which might either remain the same or increase. The sub-MIC concentration culture from this latest passage was then employed to inoculate a new series of diluted antimicrobial compounds. This iterative process was continued for a duration of 30 days, with cells from each passage being stored. Resistance was confirmed by comparing the MICs of resistant mutants against *E. coli* that had not been previously exposed to the indicated antibiotics. Bacteria showing increased MIC values to 5-FU after passage and the original strains were selected for a subsequent whole-genome sequencing analysis. A comparative genomic analysis of the resistant mutants and the original strains was performed using breseq v0.37.1 to identify SNPs.

### 4.4. Construction of Gene Knockout Strain

The *upp* gene was knocked out in *E. coli* BW25113-IncX3 using the λRed recombination system. Initially, electrocompetent cells BW25113-IncX3 were transformed with the pCasKP plasmid. After colony PCR verification, the cells were cultured in LB broth with ampicillin, induced with 0.2% L-arabinose, and made electrocompetent cells. Homologous recombination primers were designed to include 50 bp flanking sequences of the *upp* gene and sequences adjacent to the chloramphenicol resistance gene. The sequences of primers were GGTATAATCCGTCGATTTTTTTTGTGGCTGCCCCTCAAAGGAGAAAGAGTGGCTGGAGCTGCTTCG (Forward) and ACTCAAAAAAAAGCCGACTCTTAAAGTCGGCTTTAATTATTTTTATTCTCATATGAATATCCTCCTTAG (Reverse). Using pKD3 as a template, the chloramphenicol resistance gene was amplified, purified, and electroporated into a BW25113-IncX3 strain containing the pCasKP plasmid. The bacteria were plated on LB plates supplemented with an antibiotic at 30 °C. To remove the pCasKP plasmid, the colonies were incubated at 42 °C overnight and confirmed by PCR. Subsequently, the pCP20 plasmid was introduced to excise the antibiotic resistance marker, with transformants selected on LB plates supplemented with kanamycin. Finally, pCP20 was eliminated by culturing colonies at 42 °C.

### 4.5. Transcriptomic Sequencing and Analysis

*E. coli* was grown in an LB to the exponential phase and treated with 5-FU alone for 2 h. The total RNA of the samples was extracted by an EASYspin Plus kit (Aidlab, Cat No. RN2802, Beijing, China) and quantified by using a Qubit 3.0 (Thermo Fisher Scientific, Waltham, MA, USA). The RNA samples were subsequently submitted to Beijing Saimo Lily Biotechnology (Beijing, China) and sequenced by using the Illumina novaseq6000 (Illumina, San Diego, CA, USA). For library preparation, 3 μg of quality-assured RNA is used as the starting material. Ribosomal RNA is removed using the Epicentre Ribo Zero™ rRNA Removal Kit (Epicentre, Madison, WI, USA). Sequencing libraries are constructed using Illumina’s NEBNext^®^ Ultra™ Directional RNA Library Prep Kit (New England Biolabs, Ipswich, MA, USA). High-throughput sequencing is performed on the Illumina NovaSeq 6000 platform, followed by subsequent analysis of the sequencing data. RNA-seq data are processed for quality control, alignment, and statistical analysis using Bacpipe RNA-seq processing pipeline (v0.6.0). Annotations are performed using Prokka (v1.13.3). Fastp (v0.20.0) is employed for quality control of the sequencing data, and SortMeRNA (v3.0.2) is used to filter out reads derived from rRNA. Sequences are aligned to the genome sequence using BWA (v0.7.17). Finally, differential gene expression analysis of the raw data is conducted using DESeq2 (v1.20.0), and functional enrichment analysis is performed using ClusterProfiler (v3.6.0).

### 4.6. RT-PCR Assay

Following treatment with various concentrations of 5-FU, total bacterial RNA was extracted using the EASYspin Plus Bacterial RNA Rapid Extraction Kit. Genomic DNA removal and reverse transcription were carried out using the HiScript III All-in-one RT SuperMix Perfect for qPCR (Vazyme, Cat No. R333, Nanjing, China). Quantitative polymerase chain reaction (qPCR) was performed using the SYBR™ Green Master Mix (Applied Biosystems™, Cat No. A25742, Waltham, MA, USA). The transcriptional levels of *bla*_NDM-5_ relative to the internal control gene (16S ribosomal RNA) were determined. The primer sequences are listed in Appendix A.

### 4.7. 8-OHdG Level Determination

The 8-OHdG level of *E. coli* was determined by the 8-OHdG ELISA Assay Kit (Sangon Biotech, Cat No. D751009, Shanghai, China). The *E. coli* strain was incubated to the exponential phase and treated with 5-FU, and the bacterial cultures were collected, centrifuged, and resuspended. The supernatant was used to determine the levels of 8-OHdG following the manufacturer’s instructions.

### 4.8. NAD^+^/NADH Determination

The *E. coli* strain was incubated to the exponential phase and treated with 5-FU, and the bacterial cultures were collected, centrifuged, and resuspended. The NAD^+^/NADH ratio was determined by using an NAD^+^/NADH Assay Kit (Beyotime, Cat No. S0175, Shanghai, China), following the manufacturer’s instructions.

### 4.9. ATP Measurement

The intracellular ATP level of *E. coli* was determined by the Enhanced ATP Assay Kit (Beyotime, Cat No. S0027, Shanghai, China). The overnight culture of bacteria was diluted 1:100 into fresh LB and incubated at 37 °C for 4–6 h. Then, the culture was treated with different concentrations of 5-FU at 37 °C. Then, the bacterial cultures were collected, lysed, and centrifuged, and the supernatants were applied to determine the intracellular ATP level.

### 4.10. Measurement of ROS Activity

The levels of ROS in *E. coli* treated with 5-FU were measured with 10 μM of 2′,7′-dichlorofluorescein diacetate (DCFH-DA), following the manufacturer’s instruction (Beyotime, Cat No. S0033, Shanghai, China). After incubation and washing with PBS buffer, the fluorescence intensity was immediately measured with the excitation wavelength at 488 nm and the emission wavelength at 525 nm using the Infinite M200 Microplate reader.

### 4.11. Time-Dependent Killing Curves

*E. coli* 15NN1 were prepared by diluting overnight cultures 1/100 in Mueller–Hinton broth (MHB), followed by incubation for 4 h (exponential phase) or 8 h (stationary phase). The bacteria were then treated with 5-FU for 24 h at 37 °C or 0 °C. At specified time intervals, the 100 μL aliquots were removed, centrifuged, and resuspended in 100 μL sterile PBS. Subsequently, ten-fold serially diluted suspensions were plated on MHA plates and incubated overnight at 37 °C. The bacterial colonies were counted, and the primary CFUs/mL was calculated.

### 4.12. Statistical Analyses

Statistical analyses were performed using GraphPad Prism 9.5.0 (Software Inc., San diego, CA, USA). The data were presented as the mean ± SD, and one-way ANOVA with Dunnett’s multiple comparisons were used to calculate *p* values. All *p* values less than 0.05 were considered statistically significant and were indicated in the figure legends. *: *p* < 0.05; **: *p* < 0.01; ***: *p* < 0.001; ****: *p* < 0.0001.

## 5. Conclusions

In summary, we found that the *upp* gene played a pivotal role in mediating resistance to 5-FU in *E. coli* through serial passage selection of resistant strains. Further antimicrobial mechanism analysis through transcriptomic profiles and physiological trials indicated that 5-FU can not only induce bacterial DNA damage but also increase the intracellular ATP levels and produce ROS, thereby exerting its antimicrobial effects. These findings suggest that the antibacterial mechanism of 5-FU might be a multi-target mode that affects many molecular pathways in *E.coli*, which lays the groundwork for further exploitation of 5-FU as a therapeutic candidate against bacterial infections.

## Figures and Tables

**Figure 1 antibiotics-13-00528-f001:**
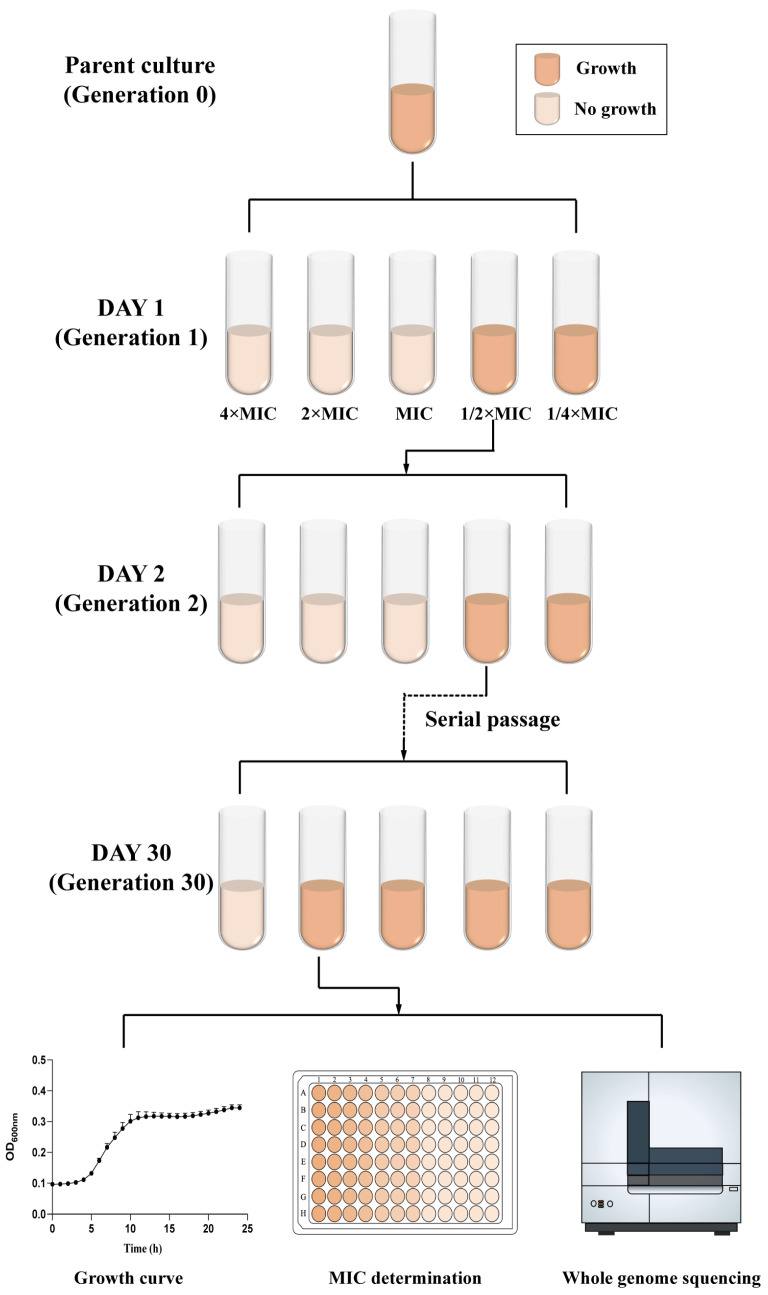
Schematic illustration of serial passage experiments for the generation of 5-FU-resistant mutant strains.

**Figure 2 antibiotics-13-00528-f002:**
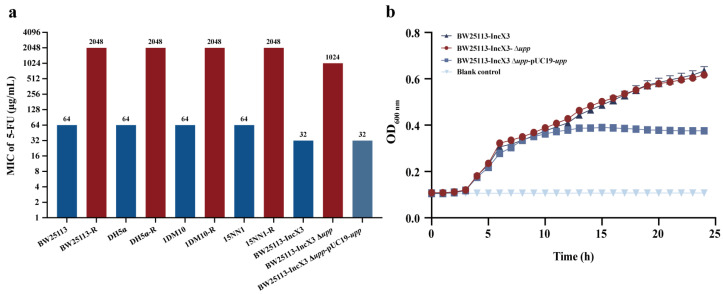
The susceptibility and growth curves of *E. coli* strains. (**a**) The MIC determination of 5-FU in *E. coli*; (**b**) The growth curves of 5-FU against three *E. coli*.

**Figure 3 antibiotics-13-00528-f003:**
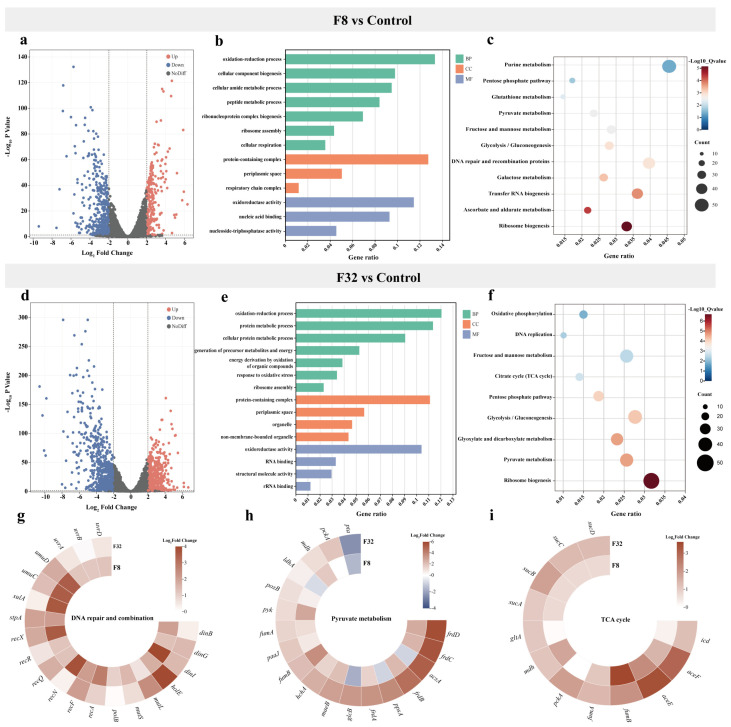
Transcription profile of *E. coli* upon co-culture with 5-FU. (**a**,**d**) Volcano diagram of DEGs; (**b**,**e**) GO biology processes of DEGs; (**c**,**f**) KEGG pathways of DEGs; (**g**–**i**) Selected DEGs involved in DNA repair and combination, pyruvate metabolism, and TCA cycle. Concentrations of 5-FU used in the experiments are 8 µg/mL (F8) and 32 µg/mL (F32), with Control indicating the untreated control group.

**Figure 4 antibiotics-13-00528-f004:**
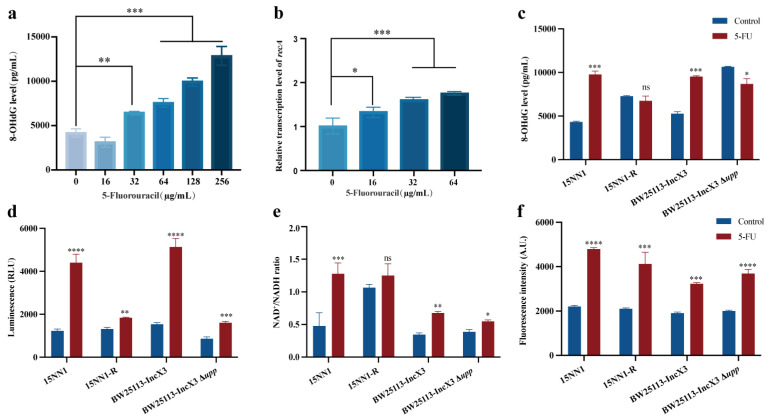
The impact of 5-FU on bacterial DNA damage and energy metabolism. (**a**) The intracellular 8-OHdG levels in *E. coli* 15NN1 treated with 5-FU. (**b**) The effect of 5-FU on the transcriptional levels of *recA*. (**c**) ELISA measurements of 8-OHdG levels in original and resistant *E. coli* treated with 5-FU (2×MIC); the concentrations of 5-FU used were 128 µg/mL and 4096 µg/mL for *E. coli* 15NN1 and 15NN1-R, respectively, and 64 µg/mL and 2048 µg/mL for BW25113-IncX3 and BW25113-IncX3 ∆*upp* strains. The determination of intracellular ATP (**d**), NAD^+^/NADH ratio (**e**), and ROS (**f**) in original and resistant *E. coli* treated with 5-FU (2×MIC). *p* value was determined using one-way ANOVA. * *p* < 0.05, ** *p* < 0.01, *** *p* < 0.001, **** *p* < 0.0001.

**Table 1 antibiotics-13-00528-t001:** SNPs in 5-FU-resistant mutant strains.

Bacteria ID	Mutation	Annotation	Description
BW25113	C→T	Q65* (CAG→TAG)	Uracil phosphoribosyltransferase
	G→A	W106* (TGG→TAG)	Melibiose operon regulatory protein MelR, AraC family
	T→A	D105V (GAT→GTT)	Thymidine kinase
DH5α	A→G	Q36Q (CAA→CAG)	c-di-GMP phosphodiesterase
	G→T	E105* (GAA→TAA)	Uracil phosphoribosyltransferase
	T→G	K13Q (AAG→CAG)	Small inner membrane protein,
15NN1	2 bp → CA	Coding (218–219/726 nt)	Uridine monophosphate kinase
	C→A	L29F (TTG→TTT)	Chromosomal replication initiator protein DnaA
	T→C	I49T (ATC→ACC)	S-adenosylmethionine decarboxylase proenzyme, prokaryotic class 1A
1DM10	T→G	K14Q (AAG→CAG)	Uracil phosphoribosyltransferase
	G→A	Q185* (CAG→TAG)	Uracil phosphoribosyltransferase
	A→C	D158E (GAT→GAG)	Phage tail fiber protein
	G→A	P182P (CCC→CCT)	Molybdenum ABC transporter permease protein ModB

## Data Availability

Data are contained within this article.

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
