# Peer review of "Deciphering the Antibacterial Mechanisms of 5-Fluorouracil in Escherichia coli through Biochemical and Transcriptomic Analyses"

_antibiotics, 2024, doi:10.3390/antibiotics13060528_

Round 1
Reviewer 1 Report
Comments and Suggestions for Authors
The manuscript "Deciphering the antibacterial mechanisms of 5-fluorouracil in Escherichia coli through biochemical and transcriptomic analyses" is devoted to elucidation of the mechanism of action of 5-fluoroacil (5-FU) using a number of genetic and biochemical assays. The manuscript is well-written and illustrated, the methods are adequately applied and described and the conclusions are supported by the results. Sequencing of resistant isolates revealed mutations in upp gene, and the fact that its loss of function mediates 5-FU resistance was further supported by a knock-out experiment. Transcriptomic analysis of bacterial cells treated with 5-FU indicates multimodal mode of action. Despite the low level of antibacterial activity of the tested compound, these findings can prove useful for further drug development.
Minor comments:
1. Figure 2a may look more informative on a logarithmic scale.
2. Probably, a comparison of bactericidal and bacteriostatic effects of 5-FU on bacterial cells in exponential and stationary phases (or even on cold-arrested dormant cells) can contribute to further elucidation of the mode of its action
Author Response
Dear reviewer,
Thank you for your comments and providing good suggestions for our manuscript. My point-by-point explanations in response to the comments are provided in the attached Word document. Please see the attachment for details.

Reviewer 2 Report
Comments and Suggestions for Authors
In the manuscript entitled “Deciphering the antibacterial mechanisms of 5- fluorouracil in Escherichia coli through biochemical and transcriptomic analyses”; M.Zhang et al, elucidated antibacterial activity of 5-flurorouracil (5FU) and found 5FU induces bacterial DNA damage and promote ROS production in bacterial cells. Authors also found mutation in upp gene increases MIC of 5- fluorouracil against E.coli. The findings are interesting and well presented. However, to make it publishable authors are requested to consider the following.
1. Present MIC data with variability in data set for the three biological replicates (Figure 2A)
2. Elaborate the methodology for the generation of resistant E.coli strains.
3. Include gene knockout verification PCR results in the supporting file.
4. Improve the quality of (Figure 3) by increasing the font size. Labels are not clear.
5. It is interesting and noteworthy that generation of ROS was quite high in upp deleted E. coli strain, which indicates 5FU is clearly toxic but the modified strain was resistant. Please comment.
6. Also, 2X MIC is a relative term and different for resistant and susceptible strains. So it is better to indicate concentration as well. Required in results indicated in Figure 4.
Author Response
Dear reviewer,
Thank you for your careful review and constructive suggestions regarding our manuscript. We have revised the manuscript in accordance with your comments and marked all the amends in the revised manuscript.
- Present MIC data with variability in data set for the three biological replicates (Figure 2A)
Response: Thanks for your suggestion. The MIC results for the three biological replicates were consistent, showing no variability. Therefore, no error bars are displayed in Figure 2A. Detailed information regarding the MIC data were presented in Table S1.
- Elaborate the methodology for the generation of resistant E. coli strains.
Response: We have revised and optimized the methodology section, and the methodology for generating resistant E. coli strains is detailed in section 4.3 in lines 268-277.
- Include gene knockout verification PCR results in the supporting file.
Response: The gene knockout verification PCR results has been added in Figure S3 in supplementary data.
- Improve the quality of (Figure 3) by increasing the font size. Labels are not clear.
Response: We have improved the quality of figure 3 by increasing the font size of all labels.
- It is interesting and noteworthy that generation of ROS was quite high in upp deleted E. coli strain, which indicates 5-FU is clearly toxic but the modified strain was resistant. Please comment.
Response: Thank you for your insightful observation. For the resistant strains, we used 5-FU at 2 × MIC (4096 µg/mL for 15NN1-R and 2048 µg/mL for BW25113-IncX3 ∆upp). At these concentrations, 5-FU is effective in inhibiting the growth of resistant strains, and the generation of ROS plays a role in this antimicrobial action.
- Also, 2 × MIC is a relative term and different for resistant and susceptible strains. So, it is better to indicate concentration as well. Required in results indicated in Figure 4.
Response: Thank you for your suggestion regarding the indication of specific concentrations corresponding to 2 × MIC for both resistant and susceptible strains. To avoid clutter in Figure 4, we have included these details in the figure legend. Specifically, the concentrations are as follows: 128 µg/mL and 4096 µg/mL for 15NN1 and 15NN1-R, and 64 µg/mL and 2048 µg/mL for BW25113-IncX3 and BW25113-IncX3 ∆upp strain.
